# The gender gap in Ph.D. entrepreneurship: How do students perceive the academic environment?

**Alessandro Muscio** [1] *, **Giovanna Vallanti** [2]

1 Dipartimento di Economia, Management e Territorio (DEMeT), Università di Foggia, Foggia, Italy,
2 Dipartimento di Economia e Finanza, Università Luiss Guido Carli, Roma, Italy

* alessandro.muscio@unifg.it

**Data Availability Statement:** The data used in this article is fully available in STATA format (.dta) and has been uploaded as a Supporting Information file.

## Abstract

This paper investigates gender issues in Ph.D. entrepreneurship. The empirical analysis is based on data from a questionnaire survey run in 2014–15 in Italy. We analyse how Ph.D. students perceive the institutional entrepreneurial environment, the drivers and the factors hindering entrepreneurship and gender-equality among faculties at the parent institution. We find evidence of a gender bias in Ph.D. entrepreneurship and that the perception about the factors either hampering or supporting entrepreneurship is deeply different between sexes. The academic environment can have a fundamental impact on students' decisions to start new ventures and on the probability that they will abandon their entrepreneurial intentions. Female student entrepreneurs particularly benefit from the opportunity to engage with a gender-balanced work environment.

## 1 Introduction

Many modern societies have set the achievement of gender equality as a top priority. However, entrepreneurship remains nowadays a male-dominated venture [1], even if women could bring different perspectives and mindsets to enterprises [2, 3], contributing enormously to the socio-economic development of nations [4]. The interest in gender issues has been growing even in contexts such as academic entrepreneurship. Academic entrepreneurship represents a fast-growing issue in the scientific literature, and yet our understanding about gender issues in this context is still limited. While women are involved more than ever in science, their engagement in academic entrepreneurship deserves some consideration [5] and a better understanding about how women entrepreneurs can be supported is needed [6]. In this respect, few recent papers have highlighted how the environment created by higher education institutions can be a key factor in promoting and supporting academic entrepreneurship [7–10].

With the rise in relevance of academic engagement in science policy, academic institutions have put increasing efforts in harvesting an entrepreneurial climate favouring entrepreneurship [11–14]. However, while universities often represent the context where scholars and students can have access the necessary knowledge and information for entrepreneurial initiatives [15], there are high levels of heterogeneity among institutions in terms of resources and

**Funding:** The authors received no specific funding for this work.

**Competing interests:** The authors have declared that no competing interests exist.

capabilities devoted to the promotion of entrepreneurship [12, 16–18]. Therefore, investigating the role played by the academic environment provides indications on how to create the appropriate conditions in academic institutions for faculty staff and students to establish entrepreneurial ventures [8].

Some papers have analysed from a gender perspective the entrepreneurial attitudes of academics and students [19–21]. Female students and academics face substantial barriers establishing business ventures and greater awareness about the reasons and causes of this gender gap in entrepreneurial activity could have tangible implications in setting university goals and practices with respect to firm creation. In fact, even if the existence of a significant gender gap in academic entrepreneurship is documented, there is still considerable controversy over the precise reasons for this gap, which in turn has implications on the design of appropriate policy responses [19].

In the attempt to fill this gap in the literature, this paper investigates the entrepreneurial activity of women among Ph.D. students in a large, entrepreneurship- oriented country such as Italy. Ph.D. entrepreneurship is a relatively novel topic in the scientific literature that is rapidly gaining credit [17, 22, 23] but gender issues among doctorate students are still relatively unexplored. Ph.D. entrepreneurship deserves special attention for several reasons. First of all, there is growing consensus among scholars regarding the fact that Ph.D. students can contribute greatly to knowledge transfer processes and are often more motivated than academics to become start- uppers since they are younger in age, less averse to risk, and in general do not have a long-term job contract. Ph.D. students may be potentially better at overcoming the barriers to new venture creation because, unlike tenured academic staff, they are often better positioned to gain access to the required commercial competences and assets, and they do not need 'genetic mutation' to become entrepreneurs [24]. Secondly, Ph.D. entrepreneurship is relevant from the policy viewpoint because it is more likely than other forms of entrepreneurship to generate knowledge-intensive start-ups, high-skilled jobs and contribute to knowledge-based regional development processes, confirming the pivotal role of universities in regenerating local communities. In fact, during their early academic careers, Ph.D. students can exploit business ideas with higher levels of technological/knowledge content than graduates and are more committed to entrepreneurial ventures than faculty members [13, 17].

Drawing from these propositions, this article aims to investigate if and in what respect women perceive differently from men universities' institutional environments. In this paper we investigate (1) the gender gap in Ph.D. entrepreneurship, (2) the relevance of the drivers and obstacles to entrepreneurship for both female and male students and (3) how the perception about the university entrepreneurial environment is associated to Ph.D. students' success in starting a business venture. For this purpose, we concentrate on two aspects of the entrepreneurial environment that Ph.D. students engage with: (a) academic efforts in creating a favourable environment for start-up creation, and (b) gender equality in the academic workplace.

The institutionalisation of entrepreneurship support, for example with initiatives such as the creation of incubators or the introduction of academic regulatory norms in favour of entrepreneurship, has been found to be positively related to the entrepreneurial climate among students [25, 26]. In this respect, many universities have created infrastructures such as business incubators in the attempt to encourage students to become entrepreneurs [27–29]. Similarly, there is evidence that academic rules for potential entrepreneurs can enhance the entrepreneurial spirit, facilitating the formation of positive perceptions among students about entrepreneurial employment outcomes [10, 12, 30] and encourage the formation of a positive attitude that is conducive to entrepreneurial intentions and their realisation [31].

Secondly, while there is the assumption that academic institutions are unbiased workplaces, few papers have recently highlighted that universities in almost every country perform poorly

in terms of gender equality [32]. In most European countries, while statistics on enrolment and graduation tend to favour women, women are underrepresented in faculty positions [33–36].

The active promotion of diversity in institutions attenuates the gender-related differences in the intensity of academic engagement [37]. The lack of women in the institutional context deprives female staff or students alike of role models or mentors that can support them in overcoming the challenges associated with engaging with industry. Therefore, unbalanced academic work environments may amplify the gender gap even in terms of students' start-up activity. Therefore, as academic institutions tend to be male-dominated work environments [38–40], female Ph.D. students will strive more than male students in establishing business ventures because they will have lower chances of engaging with same-sexed peers among academic staff [41, 42], highlighting the relevance of peer effects and homophily between individuals and their peers for entrepreneurship.

We focus the investigation of gender issues in Ph.D. entrepreneurship observing university-level factors for two reasons. First, they provide immediate policy and managerial implications for research institutions to promote entrepreneurship. Second, while there is ample evidence regarding the relationship between students' personal characteristics and entrepreneurship university-level factors have been relatively understudied [43]. In fact, while the scientific literature has investigated how universities can support academic entrepreneurship in general [44], there is little understanding regarding how the university environment is associated to students' start-up activity.

## 2 Materials and methods

The analysis of Ph.D. entrepreneurship is based on data from a questionnaire survey addressed to Ph.D. students who were enrolled in Ph.D. programmes in Italy between 2008 and 2014. The questionnaire asked general questions about students and their personal characteristics, about their study period and their level of satisfaction with the study programme, their occupational status and entrepreneurial activity.

The questionnaire survey was run with the support of CINECA, an Italian consortium of universities, research institutions and the Ministry of Education and Research (MUR). The authors designed the questionnaire, CINECA verified it and administered the survey. CINECA holds the email addresses of every academic and Ph.D. student in Italy. The survey between the end of 2014 and the beginning of 2015. CINECA contacted via email 23,500 individuals, which represent 50% of the population of doctorate students that were admitted in a Ph.D. course in the period 2008–14 in Italy. No contact information was disclosed to the authors and CINECA removed any possible reference to the identity of the interviewees.

All responses were verified by the Italian National institute of Statistics (ISTAT), which did not disclose data for students enrolled in Ph.D. courses for which the response rate was too small, to ensure confidentiality (e.g., two responses from Ph.D. programmes involving just three students). ISTAT returned to CINECA and the authors a database with 11,908 questionnaire responses. The authors dropped the cases with missing information and finalised a database with 9,062 complete responses (39% response rate). 68% of respondents completed their Ph.D. studies and 72% were employed. 6.5% of respondents had started a business, 87% of which were still active at the time of the survey.

The authors verified the goodness of the sample balance with ISTAT data. ISTAT publishes data on the population of Ph.D. graduates in Italy in yearly reports. In the 2015 report, ISTAT identified 22,469 graduates belonging to two cohorts: 2008 and 2010. We compared the distribution of responses obtained from our survey by disciplinary-scientific field (SSD) with those

presented by ISTAT and the estimated difference between the two cohorts was in all scientific areas below the 5% threshold, demonstrating the good representativeness of our sample. The Italian classification of SSD includes 14 sectors and is compliant with the one proposed by the OECD Frascati Manual [45].

The survey data was merged to data from other sources: (1) to observe university level characteristics the analysis relied on MUR data on university size, location, and research performance; (2) Information of faculty staff was obtained from a national database publicly available on the CINECA website, which allows to extrapolate the number of faculty members for all Italian universities. It also provides detailed information on gender, academic position, and the SSD for the period of analysis; (3) Finally, the authors obtained information concerning the availability of start-up regulation from university institutional websites. Information on university incubators was obtained from Netval reports [46].

# 3 Results and discussion

## 3.1 Ph.D. entrepreneurship by gender

ISTAT collected country-level data on doctorate students that obtained their Ph.D. title in 2010, 2012 and 2014. As shown in Fig 1, the number of women graduating in the three cohorts is consistently higher than men, with women representing at least 52% of the population. Most Ph.D. students in Italy receive a government scholarship and the recent cuts to university funding [47], are reflected also in the gradual decrease in the number of graduates over time. Notwithstanding gender issues, this is worrying for the future development of the Italian research and innovation system, which already underperforms other OECD countries in terms of number of scientists.

We examine Ph.D. students' career interests using ISTAT data in the three cohorts. As shown in Fig 2, confirming the results of a recent study [48], even if with different intensity over time, becoming a self-employed worker is an employment outcome that appeals to many doctorates, especially men, at least till 2014. However, the gap between man and women is consistent. Some 40% of doctorates is employed in private organisations. These results are

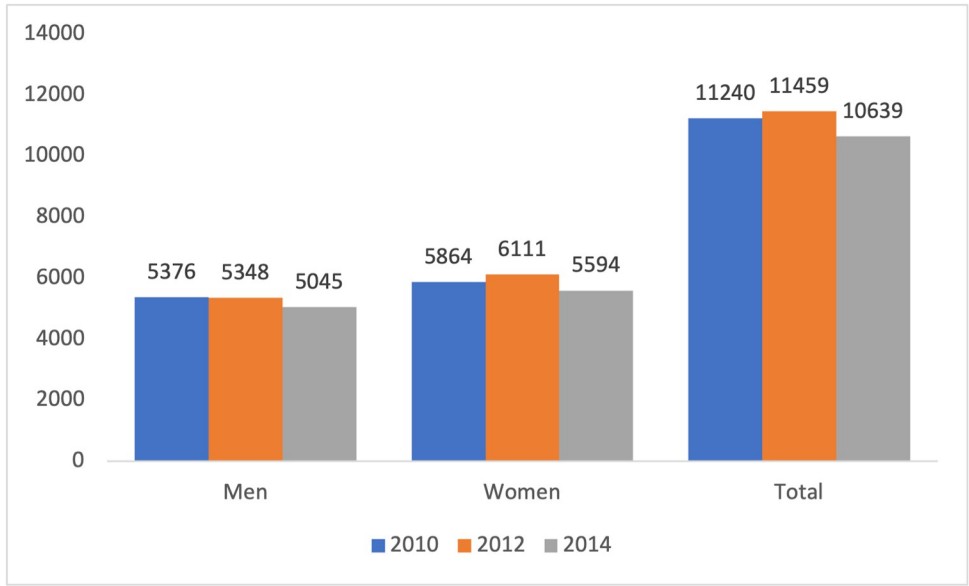

**Fig 1. Ph.D. students by gender.**

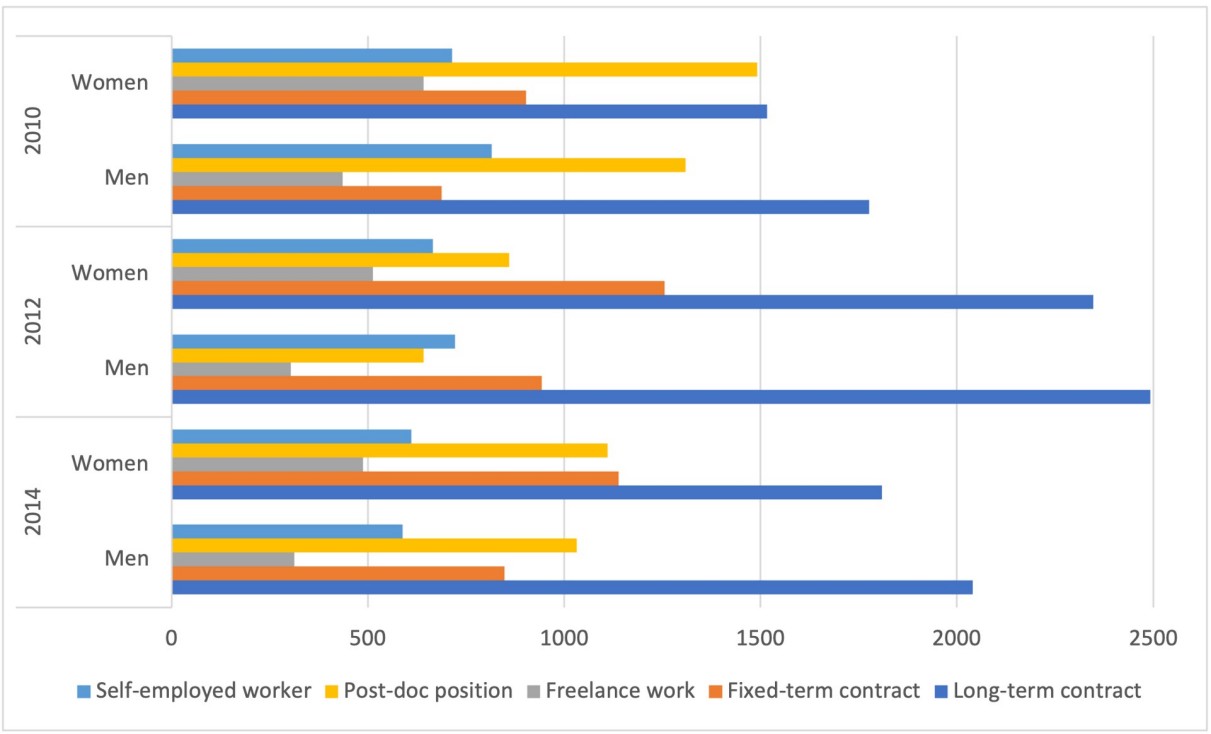

**Fig 2. Ph.D. students career interests.**

relevant from the policy perspective. Governments, especially in countries such as Italy where most students are granted a scholarship, allocate resources to Ph.D. programmes mostly on the assumption that graduates will be employed in academia, providing returns to State investments in their education [13, 49]. However, the results presented here confirm those in other works [23, 50], demonstrating that this assumption does not hold. This raises questions concerning how and to what extent universities can support students' decision to become entrepreneurs.

The analysis of survey data presented in Fig 3, show that 6.5% of Ph.D. students in Italy become entrepreneurs, demonstrating that doctorates have a much higher entrepreneurial attitude than the overall average national score of 2.8%. Looking specifically at gender differences in start-up activity, confirming ISTAT data, we find a substantial gender gap, with women being much less likely than men to achieve an entrepreneurial employment outcome (4.5% vs. 8.5%). Moreover, as shown in Fig 4, supporting recent UNESCO findings [51], in the areas of social sciences and humanities the gap between men and women in entrepreneurship is much narrower than in the hard sciences.

## 3.2 Drivers of and obstacles to entrepreneurship

In a first step we asked students about the relevance of several drivers and obstacles to entrepreneurial activity. We compared the scores of women's perception about these factors to men's perception. The degree of perception about the drivers and obstacles to entrepreneurship was measured on a Likert scale ranging from "1- absolutely not important" to "6-absolutely important". We then run independent samples t-tests to compare the difference of mean values of the responses. The following tables present the significance level expressed in

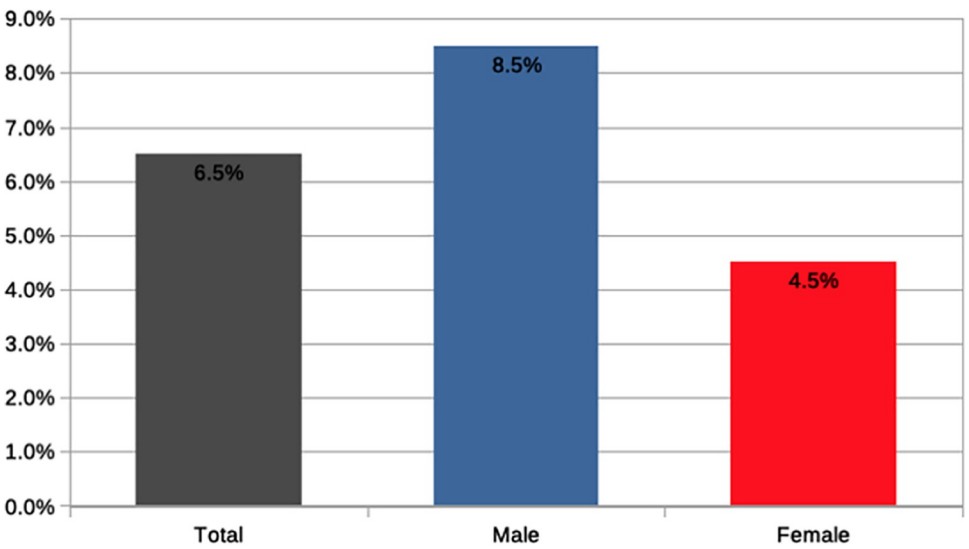

**Fig 3. Ph.D. start-up activity by gender.**

symbols. We found that the difference of mean values of the responses between males and females is different from 0 for all the factors considered in the questionnaire.

As shown in Table 1, female students have a higher perception of the relevance of the identified drivers. The difference between opinion levels is especially relevant for what concerns the access to scientific support from external peers, the availability of business competition schemes and the access to public funding. Start-up assistance and business plans are also

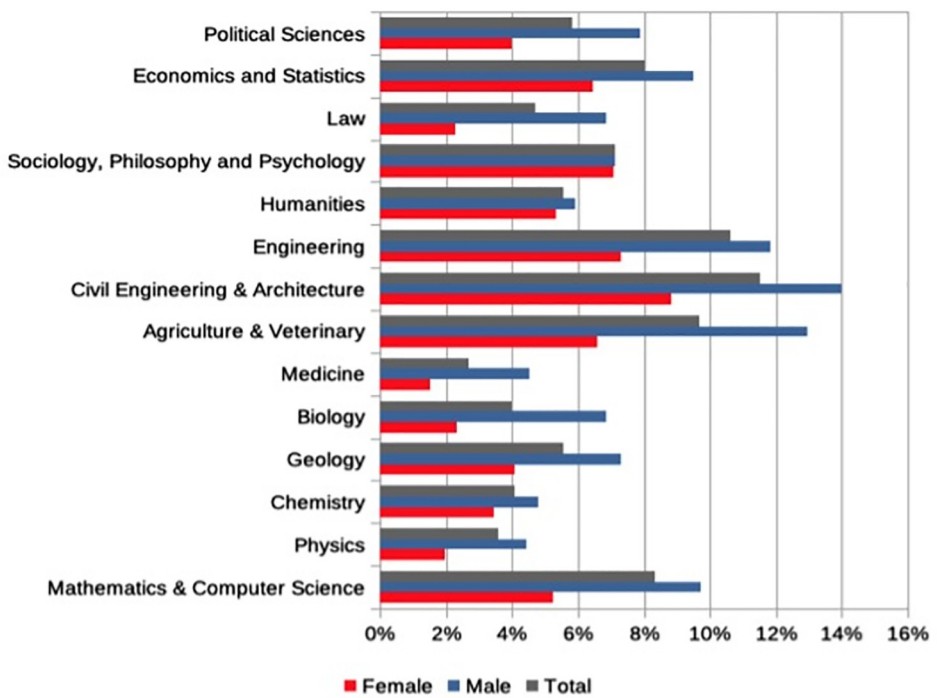

**Fig 4. Start-up activity by scientific field and gender.**

**Table 1. Students' opinion about the relevance of drivers to entrepreneurship.**

| Response | Female | Male | p-value | |
|---|---|---|---|---|
| Public funding | 4.87 | 4.53 | 0.000 | ** |
| Private funding | 4.91 | 4.73 | 0.000 | ** |
| Bank loans | 4.77 | 4.60 | 0.000 | ** |
| Financial partners | 4.87 | 4.72 | 0.000 | ** |
| Industrial partners | 4.88 | 4.82 | 0.005 | * |
| Field experts | 5.15 | 4.96 | 0.000 | ** |
| Patent portfolios | 4.15 | 3.92 | 0.000 | ** |
| Scientific support | 4.25 | 3.86 | 0.000 | ** |
| Start-up assistance | 4.90 | 4.62 | 0.000 | ** |
| Business plan | 4.84 | 4.58 | 0.000 | ** |
| Market analysis | 4.81 | 4.61 | 0.000 | ** |
| Incubation | 4.58 | 4.34 | 0.000 | ** |
| Design | 4.67 | 4.44 | 0.000 | ** |
| Legal support | 5.07 | 4.88 | 0.000 | ** |
| Business competition schemes | 4.80 | 4.46 | 0.000 | ** |

Independent samples *t*-test

* significant at 5 per cent level

** significant at 1 per cent level.

relevant. This provides insights about a greater need for women of support services to entrepreneurship and mentorship.

Secondly, in Table 2 we analysed the obstacles to entrepreneurship. Supporting what was found for the drivers, we found that, overall, women face bigger obstacles than men in establishing a business venture and, as suggested in the literature [52], practices to counteract segregation and promote gender equality in this context should be promoted. Female students are especially sensitive to difficulties in finding appropriate competencies in terms of scientific

**Table 2. Students' opinion about the relevance of obstacles to entrepreneurship.**

| Response | Female | Male | p-value | |
|---|---|---|---|---|
| Unclear (lack of) academic rules and guidelines on start-up creation | 3.14 | 3.04 | .000 | ** |
| Difficulties in raising financial resources | 4.26 | 4.17 | .000 | ** |
| Difficulties in finding appropriate know-how and scientific competencies | 3.70 | 3.51 | .000 | ** |
| Difficulties in finding appropriate managerial competencies | 3.90 | 3.72 | .000 | ** |
| Difficulties in finding appropriate equipment and capital goods | 3.71 | 3.48 | .000 | ** |
| Difficulties in finding information on markets | 3.51 | 3.47 | .400 | |
| Lack of networks | 3.74 | 3.67 | .015 | ** |
| Low risk attitude | 3.46 | 3.51 | .035 | * |
| Low entrepreneurial attitude of the supervisor | 3.47 | 3.33 | .000 | ** |
| Lack of suitable partners | 3.59 | 3.48 | .000 | ** |
| Excessive bureaucracy | 4.31 | 4.34 | .251 | |
| Necessity of authorisations | 4.06 | 3.95 | .000 | ** |

Independent samples *t*-test

* significant at 5 per cent level

** significant at 1 per cent level.

Table 3. Students' opinion about the entrepreneurship environment at the parent university.

| Response | Female | Male | p-value | |
|---|---|---|---|---|
| There is a favourable environment to start-up creation | 2.86 | 2.98 | .000 | ** |
| There is a favourable environment to U-I interaction | 3.13 | 3.20 | .004 | * |
| Entrepreneurship is a central mission | 2.73 | 2.77 | .067 | |
| There is support to patenting and innovation | 3.00 | 3.06 | .012 | |
| There is dedicated strategy for technology transfer | 2.75 | 2.85 | .000 | ** |
| U-I collaboration is important | 3.27 | 3.42 | .000 | ** |
| Teaching is well connected to research | 3.56 | 3.56 | .821 | |
| There are training courses for entrepreneurs | 2.58 | 2.73 | .000 | ** |
| There is professional support to potential entrepreneurs | 2.57 | 2.76 | .000 | ** |

Independent samples *t*-test

* significant at 5 per cent level

** significant at 1 per cent level.

and managerial know-how, as well as in finding appropriate equipment and capital goods. Their risk-aversion also seem to be lower than in the case of men.

As the environment created by higher education institutions can influence greatly academic entrepreneurship [13, 53], it is important to determine how universities can create the appropriate conditions for faculty staff and students to establish entrepreneurial ventures [8]. In this line of thinking, we asked students' opinion about the entrepreneurship environment at the parent university (Table 3). Overall, women have a lower opinion about the entrepreneurial environment available to them at the parent institution than men. All scores were lower for female students than for men. If we focus on responses that were statistically different, we find that female students find universities less supporting in providing professional support to potential entrepreneurs, in offering training courses for entrepreneurs and in creating collaboration agreements with private companies. This once again supports the implementation of dedicated services to women's entrepreneurial activity. Practices to reduce this gap in students' perception must be introduced in order to promote entrepreneurship and express untapped potential and skills.

### 3.3 University initiatives for start-up activity

Many universities have started promoting the creation of an institutional environment that is favourable to business creation [11, 12, 14]. The environment that academic institutions create can be a key determinant of student and faculty entrepreneurship [7–10].

Several institutions have established business incubators in the attempt to encourage students to become entrepreneurs [11, 27, 29]. University incubators have a positive impact on start-up activity, survival, and growth [26], offering basic business support services, mentoring in early business stages and access to funding. It follows that the availability of business incubators has been typically associated in the literature to higher levels of academic entrepreneurship [11, 15].

In the attempt to creating a favourable environment to entrepreneurship, some universities have also introduced dedicated regulatory frameworks to frame start-up activities [10, 12, 30]. These norms for start-up and spin-off creation set the distribution of financial returns from entrepreneurial ventures, define the commitments of future entrepreneurs with the parent university limiting potential conflicts [12]. The adoption of these rules has a positive impact on academic entrepreneurship [54–56].

According to this line of arguments, the institutionalisation of entrepreneurship support, either via incubators or regulatory norms, should be positively related to the entrepreneurial climate among students [25]. These academic initiatives can enhance the entrepreneurial spirit, facilitating the formation of positive perceptions among students about entrepreneurial employment outcomes and encouraging the formation of a positive attitude that is conducive to entrepreneurial intentions and their realisation [31].

Notwithstanding the support that students can receive from an entrepreneurship- friendly academic environment, there is no evidence concerning any gender-specific effect of the academic institutional environment. In other words, entrepreneurship norms or support facilities should work equally well (or bad) for both men and women. Supporting this, some authors [57] analyse the impact of non-financial support policies and programmes, such as consulting and monitoring services to early-stage companies, on start-up activity, finding no significant variation between different sexes. In this sense, the needs and problems of women entrepreneurs that institutions can address, should not be too different from those of men [58, 59].

Table 4 reports information on gender differences in Ph.D. students' entrepreneurial attitude as well three dimensions of the university entrepreneurial environment available to students engaged in a Ph.D. programme: the availability of university polices for entrepreneurship, the availability of an incubator and the share of women in faculty, in the same scientific area and in total. Information concerning university policies such as the availability of an incubator or start-up regulation was obtained from institutional websites. Information on staff was drawn from ministerial data. As shown in Table 4, considering gender differences in entrepreneurial intentions and activity, we find a substantial and statistically significant gender gap in both cases, with women exhibiting a lower attitude towards entrepreneurship and being less likely to start a new venture. We find also that university initiatives do not have a gendered association to men or women. However, the statistics reported in Table 4 suggest a gender self-selection in the Ph.D. programme, with women being engaged more than men in Ph.D. programmes offered in universities and scientific areas where the share of female faculty is relatively higher.

Whether the opportunity of working in a more gender balanced academic environment also affects female attitude towards entrepreneurship is investigated by means of a logit regression model as reported in Table 6. The logit specification allows to estimate the gender

**Table 4. The university environment of student entrepreneurs.**

| Response | Female | Male | p-value | |
|---|---|---|---|---|
| Rate of entrepreneurship (1) | 4.5% | 8.5% | .000 | ** |
| Entrepreneurial intention (1 = no int. ~ 6 = high int.) (2) | 3.70 | 3.95 | .000 | ** |
| Availability of a dedicated start-up and spinoff regulation (Y/N) | 0.64 | 0.62 | .028 | * |
| Availability of a business incubator (Y/N) | 0.81 | 0.81 | .578 | |
| Share of university female faculty in the same year of graduation and area of the Ph.D. student | 0.38 | 0.33 | .000 | ** |
| Share of university female full professors in the same year of graduation and area of the Ph.D. student | 0.23 | 0.19 | .000 | ** |

Independent samples t-test

* significant at 5 per cent level

**significant at 1 per cent level.

(1) Percentage of students who established or contributed to the establishment of a business start-up

(2) Students' intention in creating a start-up measured on a Likert scale ranging from "1—no intention" to "6—extremely high intention".

**Table 5. Description of variables.**

| Variable | Description | Data source |
|---|---|---|
| Start-up | Dummy variable taking the value 1 if the student established or contributed to the establishment of a business start-up and 0 otherwise. | Questionnaire |
| Start-up intention | Dummy variable taking the value 1 if the student intends to create a business start-up and 0 otherwise. Start-up intention was measured in the questionnaire on a Likert scale ranging from 1 to 6 = highest. The dummy variable is equal to 1 for responses > 4 and 0 otherwise. | Questionnaire |
| Female | Dummy variable taking the value 1 if the student is female and 0 otherwise. | Questionnaire |
| Age start-up regulation | Age of dedicated policies for spinoff and start-up creation (in 2006, if available). | University website |
| Business incubator | Dummy variable taking the value 1 if a business incubator is available at the parent institution. | PniCube website |
| Share female staff | Share of university female faculty in the same year of graduation of the Ph.D. the student and in the same scientific area. | MUR |
| Age | Age of the student. | Questionnaire |
| Entrepreneur parent | Dummy variable taking the value 1 if at least one of the student's parents is an entrepreneur. | Questionnaire |
| Academic position | Dummy variable taking the value 1 if the student holds an academic position. | Questionnaire |
| No work experience | Dummy variable taking value 1 if the student had no job experience before the beginning of the Ph.D. programme, and 0 otherwise. | Questionnaire |
| Risk preference | Scalar variable ranging from 1 if the student claims that she/he is more willing to invest in technologies, projects or products that involve low risk and certain, low gains and 5 if she/he is more willing to invest in risky projects that involve high gains. | Questionnaire |
| University rating | Research rating published by MUR in 2014, based on evaluation of the research output carried out over the period 2004–10. This composite indicator accounts for peer review evaluations of research activity carried out at academic institutions (patents, impact factor of journal articles, etc.). | MUR |
| University size | University size is expressed as numbers of students: 1 small (<10,000); 2 medium (10,000–15,000); 3 large (15,000–40,000); 4 mega (>40,000). | MUR |
| Unemployment rate | Unemployment rate in the province (NUTS3) where the university is located, in the year before graduation. | ISTAT |

differences in entrepreneurship (both entrepreneurial intention and actual start-up activity) among Ph.D. students in relation to the university entrepreneurial environment by controlling for several individual characteristics which may affect differently the entrepreneurial attitude of male and female students, such as risk attitude and parents' entrepreneurship and work experience before starting the Ph.D., along with some university characteristics. Table 5 presents the description of variables used in the econometric analysis. In Table 6, columns (1) and (4) present the results including the whole sample of respondents, while columns (2) and (3) for start-up and (5) and (6) for entrepreneurship intention present the results separately for the two samples of female and male students.

Confirming the results of previous literature, after controlling for individual and university characteristics, we find that university policies for entrepreneurship are equally important for men and women. However, what really seems to matter for women's success in starting a business is the opportunity to engage during their study period with same-sexed role models. Our results suggest that those women who attended universities where the faculty gender balance was more favourable to women, were more likely to successfully establish a business venture, while the effect on men is negligible and not statistically significant. These results extend also to entrepreneurial intentions.

These findings suggest that attending institutions where the faculty is more gender- balanced provides great incentives to women to become entrepreneurs, without necessarily having negative effects on men. This implies that women benefit from engaging with same-sexed role models and being in environments where women are rightly represented allows them to feel more confident in overcoming gender barriers and venture into start-up prospects. In contrast, men do not seem to have the same need for same-gender role models [41] as the rate of entrepreneurship is largely in their favour. Unfortunately, female Ph.D. students engage with

**Table 6. Logit regressions.**

| VARIABLES | Start-up (1) | Start-up F (2) | Start-up M (3) | Start-up intention (4) | Start-up intention F (5) | Start-up intention M (6) |
|---|---|---|---|---|---|---|
| Female | -0.030** | - | - | -0.042** | - | - |
| | [0.007] | | | [0.011] | | |
| Age start-up regulation | 0.003* | 0.002 | 0.004+ | 0 | -0.004 | 0.004 |
| | [0.001] | [0.002] | [0.002] | [0.002] | [0.003] | [0.003] |
| Business incubator | 0.031** | 0.024* | 0.041** | 0.039* | 0.062** | 0.02 |
| | [0.010] | [0.012] | [0.016] | [0.016] | [0.024] | [0.021] |
| Share female faculty | 0.072+ | 0.080+ | 0.051 | 0.200** | 0.330** | 0.073 |
| | [0.040] | [0.047] | [0.066] | [0.073] | [0.112] | [0.097] |
| Age | 0.002** | 0.002** | 0.003* | 0.000 | 0.000 | 0.000 |
| | [0.001] | [0.001] | [0.001] | [0.001] | [0.002] | [0.002] |
| Entrepreneur parent | 0.050** | 0.041** | 0.059** | 0.057** | 0.032 | 0.089** |
| | [0.009] | [0.011] | [0.016] | [0.018] | [0.025] | [0.026] |
| Academic position | -0.020** | -0.021* | -0.020+ | -0.066** | -0.067** | -0.063** |
| | [0.007] | [0.009] | [0.011] | [0.012] | [0.018] | [0.015] |
| No work experience | -0.025** | -0.023* | -0.029* | -0.007 | -0.011 | -0.004 |
| | [0.007] | [0.009] | [0.012] | [0.012] | [0.018] | [0.016] |
| Risk preference | 0.020** | 0.015** | 0.026** | 0.081** | 0.082** | 0.080** |
| | [0.004] | [0.006] | [0.007] | [0.007] | [0.011] | [0.009] |
| University rating | -0.024 | -0.014 | -0.039 | -0.031 | -0.053 | -0.017 |
| | [0.016] | [0.020] | [0.026] | [0.027] | [0.041] | [0.034] |
| University size | -0.019** | -0.014* | -0.025** | -0.018* | -0.027* | -0.01 |
| | [0.004] | [0.005] | [0.007] | [0.008] | [0.012] | [0.010] |
| Unemployment rate | 0.001 | 0 | 0.002 | 0.005** | 0.005* | 0.004* |
| | [0.001] | [0.001] | [0.001] | [0.001] | [0.002] | [0.002] |
| Constant | yes | yes | yes | yes | yes | yes |
| Area dummies | yes | yes | yes | yes | yes | yes |
| Year dummies | yes | yes | yes | yes | yes | yes |
| Observations | 5,984 | 2,855 | 3,101 | 6,400 | 3,100 | 3,300 |

+ significant at 10 per cent level

* significant at 5 per cent level

**significant at 1 per cent level.

biased environments where, notwithstanding their good academic performance, female academics are poorly represented [38–40]. Therefore, they lack the necessary peer effects that seem to work in favour of entrepreneurship.

## 4 Conclusions

Ph.D. entrepreneurship is a rising topic in the scientific literature. The results of a large survey on Ph.D. students carried out in Italy show the existence of persistent gender effects in students' perception about the drivers and obstacles to entrepreneurship. Women are generally more inclined to perceive drivers to entrepreneurship as being more fundamental to determine start-up success than men. However, they also have a more negative perception about the obstacles to start-up creation. Our qualitative analysis also confirms that there are relevant differences between men and women about how they perceive the support they receive from their parent institutions, raising concerns about the effectiveness of policies in support of the

creation of an entrepreneurial university. Overall, men are more aware than women about the entrepreneurial environment available to them.

The main issue investigated in this paper is therefore very much policy oriented and of practical nature: if there is a gender gap in Ph.D. entrepreneurship, what are the university factors of the entrepreneurial environment available at parent institutions that environment mitigate this gap? In this respect, our simple statistical exercises investigate whether the academic entrepreneurial environment is associated to Ph.D. students' success in starting a business venture and to their entrepreneurial intentions. We find that, while university policies can support to some extent Ph.D. entrepreneurship, there are no significant statistical differences between men and women that either had the intention to start a company or eventually were successful in doing so, in terms of availability of university policies or incubating infrastructures.

What really seems to push women to become entrepreneurs, is the opportunity of engaging with a gender-balanced environment during the Ph.D. programme, having the opportunity to be guided and inspired by same-sexed peers. These peer effects could mitigate gender differences and reduce the gender gap in start-up activity.

These findings are important in terms of research policy. While universities have been encouraged to adopt the 'entrepreneurial university' model, emphasizing knowledge transfer and business creation, our results suggest that traditional initiatives such as creating incubators or providing university guidelines for business creation can be a necessary but not sufficient condition in reducing gender disparities. What really matters in supporting women in catching up with men in entrepreneurial activity is bringing equality in academic workplaces, granting to women equal opportunities of recruitment and career advancements. This is not just an issue of justice, but also of economic opportunity, promoting women entrepreneurship, without necessarily being detrimental to men.

## Supporting information

**S1 Data.**
(DTA)

**S1 Questionnaire.**
(DOCX)

## Acknowledgments

This work benefited from valuable input given by Davide Quaglione and Elisa Giuliani. The authors are grateful to Ugo Rizzo for his help with the data cleaning process, and to CINECA for the questionnaire administration and data collection.

## Author Contributions

**Data curation:** Alessandro Muscio.

**Writing – original draft:** Alessandro Muscio.

**Writing – review & editing:** Giovanna Vallanti.

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
