## [Decision Letter · Decision Letter 0]

27 Sep 2021

PONE-D-21-24678The Gender Gap in Ph.D. Entrepreneurship: How do students perceive the academic environment?PLOS ONE

Dear Dr. Muscio,

Thank you for submitting your manuscript to PLOS ONE. After careful consideration, we feel that it has merit but does not fully meet PLOS ONE’s publication criteria as it currently stands. Therefore, we invite you to submit a revised version of the manuscript that addresses the points raised during the review process.

We look forward to receiving your revised manuscript.

Kind regards,

Isabel Novo-Cortí

Academic Editor

PLOS ONE

Additional Editor Comments:

This paper analyses the gender differences in the university environment at the doctoral level, with data relating to Italy. Although the topic is interesting, especially in its application to entrepreneurship, both reviewers indicate the need to deepen the description of the sample and complete the analyses carried out and provide more exhaustive explanations and interpretations. For this reason, a thorough review of the work is required.

Journal Requirements:

At this time, please address the following queries

4. We note you have included a table to which you do not refer in the text of your manuscript. Please ensure that you refer to Table 2 and 3 in your text; if accepted, production will need this reference to link the reader to the Table.

Reviewers' comments:

Reviewer's Responses to Questions

**Comments to the Author**

1. Is the manuscript technically sound, and do the data support the conclusions?

Reviewer #1: Yes

Reviewer #2: Partly

2. Has the statistical analysis been performed appropriately and rigorously? 

Reviewer #1: Yes

Reviewer #2: No

3. Have the authors made all data underlying the findings in their manuscript fully available?

Reviewer #1: No

Reviewer #2: No

4. Is the manuscript presented in an intelligible fashion and written in standard English?

Reviewer #1: Yes

Reviewer #2: Yes

5. Review Comments to the Author

Reviewer #1: This paper is interesting and well written. The topic is very well focused, and the possible applications of the conclusions could be valuable. The statistic analysis is also interesting, but I would like to use a deeper analysis with a complementary method, for example, an ANOVA comparing results. If possible, I recommend completing the paper. Besides, the complements of qualitative information will substantially improve if more explicative information is provided.

Reviewer #2: Novo-Cortí & Muscio (2021) examine gender gaps in Ph.D. entrepreneurship through a 2014-15 survey of Italian doctoral students. They make comparisons between male and female students’ perceptions about successful entrepreneurship and draw connections to the support offered by their universities. Although their survey provides important data on doctoral students’ perceptions of entrepreneurship, more detail is necessary in the paper and the analyses performed to reach their conclusion that gender differences in doctoral students’ perceived university support of entrepreneurship leads to the gender gap in doctoral entrepreneurship.

First off, I commend the authors for pursuing this survey for a large, representative sample of Italian doctoral students. There is limited data on doctoral students and their preferences for entrepreneurship, so this survey will likely be of use for many topics of interest. As such, I expected more detail about the survey in this paper. In their submission, the authors reference another one of their papers that discussed the survey and provided a data sample; this information would have also been helpful in this paper for readers to get a full picture of what the survey entailed. The authors also provided an original copy of the survey in Italian with their supporting materials; I would encourage submitting an English translation of the survey as well. This would better allow English-speaking readers to follow specific survey questions as they are referenced in the paper.

Giving the readers more information about the survey will allow them to better follow the paper’s analyses, which could be performed more rigorously and with more detail. As the analyses currently stand, the results provided are lacking information to reach the conclusions drawn in the paper. The strongest evidence provided are the t-tests for gender differences in perceptions of successful entrepreneurship, obstacles, and their parent university’s entrepreneurship environment (Tables 1-3). However, for these tables, the authors did not provide a p-value and rather used asterisks to denote the range of potential p-values. Providing the actual p-value or a standard error for these t-tests could better help readers determine the significance of these results.

The authors then draw comparisons between the gender gap in entrepreneurship and entrepreneurial intention with students’ university environment. They again use a battery of t-tests; because they are drawing comparing across gender and across university characteristics, the t-tests are difficult to interpret as correlations between the two. It would be more appropriate to use a regression that interacts characteristics of the university environment with gender. The authors also describe analyses that provide different cuts of the data, such as focusing on only students with high entrepreneurial intention. However, their paper does not provide results with those separate cuts of the data; for example, Table 4 (the final table) appears to perform analyses for the entire dataset. This makes it difficult for readers to determine whether the authors’ conclusions are adequately supported by the data.

Overall, Novo-Cortí & Muscio (2021) has the potential to be informative of the gender gap in doctoral entrepreneurship. In its current state, the paper requires more detail and additional analyses to solidify the conclusions about how university support differentially affects male and female students’ entrepreneurial tendencies. Thus, I recommend a major revision and resubmission.

6. PLOS authors have the option to publish the peer review history of their article (what does this mean?). If published, this will include your full peer review and any attached files.

Reviewer #1: No

Reviewer #2: No

---

## [Author Response · Author response to Decision Letter 0]

20 Nov 2021

RESPONSE TO REVIEWERS

Reviewer #1: This paper is interesting and well written. The topic is very well focused, and the possible applications of the conclusions could be valuable. The statistic analysis is also interesting, but I would like to use a deeper analysis with a complementary method, for example, an ANOVA comparing results. If possible, I recommend completing the paper. Besides, the complements of qualitative information will substantially improve if more explicative information is provided.

RESPONSE: Thank you for the encouraging words. We did not run the ANOVA as we analyse just two groups (male vs. female). However, as part of our efforts to respond to Reviewer #2, we added an econometric analysis to the results. We hope that this responds also to your request to complete the analysis. We also added better explicative information.

Reviewer #2: Novo-Cortí & Muscio (2021) examine gender gaps in Ph.D. entrepreneurship through a 2014-15 survey of Italian doctoral students. They make comparisons between male and female students’ perceptions about successful entrepreneurship and draw connections to the support offered by their universities. Although their survey provides important data on doctoral students’ perceptions of entrepreneurship, more detail is necessary in the paper and the analyses performed to reach their conclusion that gender differences in doctoral students’ perceived university support of entrepreneurship leads to the gender gap in doctoral entrepreneurship.

First off, I commend the authors for pursuing this survey for a large, representative sample of Italian doctoral students. There is limited data on doctoral students and their preferences for entrepreneurship, so this survey will likely be of use for many topics of interest. As such, I expected more detail about the survey in this paper. In their submission, the authors reference another one of their papers that discussed the survey and provided a data sample; this information would have also been helpful in this paper for readers to get a full picture of what the survey entailed. 

RESPONSE: Thank you for the encouraging words. We extended the description of the survey.

The authors also provided an original copy of the survey in Italian with their supporting materials; I would encourage submitting an English translation of the survey as well. This would better allow English-speaking readers to follow specific survey questions as they are referenced in the paper.

RESPONSE: We translated the questionnaire and attached it to the paper.

Giving the readers more information about the survey will allow them to better follow the paper’s analyses, which could be performed more rigorously and with more detail. As the analyses currently stand, the results provided are lacking information to reach the conclusions drawn in the paper. The strongest evidence provided are the t-tests for gender differences in perceptions of successful entrepreneurship, obstacles, and their parent university’s entrepreneurship environment (Tables 1-3). However, for these tables, the authors did not provide a p-value and rather used asterisks to denote the range of potential p-values. Providing the actual p-value or a standard error for these t-tests could better help readers determine the significance of these results.

RESPONSE: We added the standard error to the tests.

The authors then draw comparisons between the gender gap in entrepreneurship and entrepreneurial intention with students’ university environment. They again use a battery of t-tests; because they are drawing comparing across gender and across university characteristics, the t-tests are difficult to interpret as correlations between the two. It would be more appropriate to use a regression that interacts characteristics of the university environment with gender. The authors also describe analyses that provide different cuts of the data, such as focusing on only students with high entrepreneurial intention. However, their paper does not provide results with those separate cuts of the data; for example, Table 4 (the final table) appears to perform analyses for the entire dataset. This makes it difficult for readers to determine whether the authors’ conclusions are adequately supported by the data.

RESPONSE: We added an econometric analysis to the paper, on the whole sample as well as on two subsamples, one including men and one including women. The results support the conclusions in the paper. We omitted the descriptive statistics because of the strict requirements of the journal but they are available upon request.

Overall, Novo-Cortí & Muscio (2021) has the potential to be informative of the gender gap in doctoral entrepreneurship. In its current state, the paper requires more detail and additional analyses to solidify the conclusions about how university support differentially affects male and female students’ entrepreneurial tendencies. Thus, I recommend a major revision and resubmission.

---

## [Decision Letter · Decision Letter 1]

3 Dec 2021

The Gender Gap in Ph.D. Entrepreneurship: How do students perceive the academic environment?

PONE-D-21-24678R1

Dear Dr. Muscio,

We’re pleased to inform you that your manuscript has been judged scientifically suitable for publication and will be formally accepted for publication once it meets all outstanding technical requirements.

Kind regards,

Isabel Novo-Cortí

Academic Editor

PLOS ONE

Additional Editor Comments (optional):

Reviewers' comments:

Reviewer's Responses to Questions

**Comments to the Author**

1. If the authors have adequately addressed your comments raised in a previous round of review and you feel that this manuscript is now acceptable for publication, you may indicate that here to bypass the “Comments to the Author” section, enter your conflict of interest statement in the “Confidential to Editor” section, and submit your "Accept" recommendation.

Reviewer #2: All comments have been addressed

2. Is the manuscript technically sound, and do the data support the conclusions?

Reviewer #2: Yes

3. Has the statistical analysis been performed appropriately and rigorously? 

Reviewer #2: Yes

4. Have the authors made all data underlying the findings in their manuscript fully available?

Reviewer #2: Yes

5. Is the manuscript presented in an intelligible fashion and written in standard English?

Reviewer #2: Yes

6. Review Comments to the Author

Reviewer #2: All of my comments from the previous submission have been addressed. I believe the edits strengthen the paper's conclusions and can thus be accepted for publication.

7. PLOS authors have the option to publish the peer review history of their article (what does this mean?). If published, this will include your full peer review and any attached files.

Reviewer #2: No

---

## [Editor Report · Acceptance letter]

19 Dec 2021

PONE-D-21-24678R1 

The Gender Gap in Ph.D. Entrepreneurship: How do students perceive the academic environment? 

Dear Dr. Muscio:

I'm pleased to inform you that your manuscript has been deemed suitable for publication in PLOS ONE. Congratulations! Your manuscript is now with our production department. 

Kind regards, 

on behalf of

Dr. Isabel Novo-Cortí 

Academic Editor

PLOS ONE